# A Model Depicting the Retail Food Environment and Customer Interactions: Components, Outcomes, and Future Directions

**DOI:** 10.3390/ijerph17207591

**Published:** 2020-10-19

**Authors:** Megan R. Winkler, Shannon N. Zenk, Barbara Baquero, Elizabeth Anderson Steeves, Sheila E. Fleischhacker, Joel Gittelsohn, Lucia A Leone, Elizabeth F. Racine

**Affiliations:** 1Division of Epidemiology and Community Health, University of Minnesota School of Public Health, Minneapolis, MN 55455, USA; mwinkler@umn.edu; 2Department of Population Health Nursing Science, University of Illinois Chicago, Chicago, IL 60612, USA; szenk@uic.edu; 3Department of Health Services, University of Washington School of Public Health, Seattle, WA 98198, USA; Bbaquero@uw.edu; 4Department of Nutrition, University of Tennessee, Knoxville, TN 37996, USA; Eander24@utk.edu; 5Law Center, Georgetown University, Washington, DC 20001, USA; sef80@georgetown.edu; 6Center for Human Nutrition, Johns Hopkins Bloomberg School of Public Health, Baltimore, MD 21205, USA; jgittel1@jhu.edu; 7Department of Community Health and Health Behavior, School of Public Health and Health Professions, University at Buffalo, Buffalo, NY 14214, USA; lucialeo@buffalo.edu; 8Department of Public Health Sciences, University of North Carolina, Charlotte, NC 28223, USA

**Keywords:** grocery store, restaurant, environment, retail, food purchasing behavior, dietary intake

## Abstract

The retail food environment (RFE) has important implications for dietary intake and health, and dramatic changes in RFEs have been observed over the past few decades and years. Prior conceptual models of the RFE and its relationships with health and behavior have played an important role in guiding research; yet, the convergence of RFE changes and scientific advances in the field suggest the time is ripe to revisit this conceptualization. In this paper, we propose the Retail Food Environment and Customer Interaction Model to convey the evolving variety of factors and relationships that convene to influence food choice at the point of purchase. The model details specific components of the RFE, including business approaches, actors, sources, and the customer retail experience; describes individual, interpersonal, and household characteristics that affect customer purchasing; highlights the macro-level contexts (e.g., communities and nations) in which the RFE and customers behave; and addresses the wide-ranging outcomes produced by RFEs and customers, including: population health, food security, food justice, environmental sustainability, and business sustainability. We believe the proposed conceptualization helps to (1) provide broad implications for future research and (2) further highlight the need for transdisciplinary collaborations to ultimately improve a range of critical population outcomes.

## 1. Introduction

Dramatic changes in the retail food environment (RFE) are evident over the past few decades, and even the past few years [1,2]. The number of traditional supermarkets are declining, while alternative grocery formats such as discount and convenience focused grocers are proliferating [1]. Food is increasingly found everywhere, across stores and businesses that are not traditionally considered “food” outlets [3,4]. Exponential growth in the number of dollar stores, pharmacies, and their grocery offerings exemplifies both of these trends [1,2,5]. Due in part to technological advances, online grocery shopping with delivery or curbside pick-up may be the wave of the future, further accelerated by consumer and federal responses (e.g., expanding online shopping options for US Department of Agriculture Supplemental Nutrition Assistance Program (SNAP) participants) to the coronavirus pandemic [6]. Still, prior to the pandemic, the majority of the American food dollar went to food prepared away from home [7]. Prepared food delivery has surged, with digital ordering and third-party delivery services helping to fuel its rise [8,9]. These changes partially reflect a growing consumer demand for convenience due to time scarcity [10,11,12], but also the decisions of a variety of other food actors including outlet owners, suppliers, and manufacturers to compete for customers through facilitating convenience. The RFE—including these recent trends—has implications for health, but also for other outcomes such as community and economic development.

Over the past 15 years, conceptual models of the RFE have played an important role in guiding research and intervention efforts, and thus have advanced the field. In 2005, the Model of Community Nutrition Environments by Glanz and colleagues identified several key components of the RFE, such as the “consumer” and “community” nutrition environments, which facilitated communication in the field [13]. The ecological framework depicting multilevel, interacting influences on what people eat by Story and colleagues positioned retail food sources as a key aspect of the physical environment [14]. In her book, Morland expanded on the pathways by which the RFE affects obesity and personal factors that moderate these relationships [15]. Yet, the recent convergence of changes in the RFE and advances in the field suggest the time is ripe to revisit how we conceptualize the RFE. Previous models tend to miss important components of the current and emerging environment, such as the wide varieties of retail food sources, involved actors, and business models, focus solely on diet and/or health as the outcomes of interest, and underemphasize the broader context that influences and interacts with the RFE to affect a diverse range of population outcomes.

The proposed model in this paper was prepared by The Healthy Food Retail Working Group leadership team. The Healthy Food Retail Working Group is a US collaboration of over 150 researchers and stakeholders jointly supported by Healthy Eating Research, a national program of the Robert Wood Johnson Foundation, and the Nutrition and Obesity Policy Research and Evaluation Network (NOPREN), which is supported by a cooperative agreement from the Centers for Disease Control and Prevention’s Division of Nutrition, Physical Activity, and Obesity. The Healthy Food Retail Working Group holds bimonthly webinars on retail food topics and convenes smaller sub-groups to explore topics in further depth and develop collaborative research, practice, or policy projects.

In March 2019, the working group leadership met at the annual NOPREN meeting and strategized on research needs and future directions including a conceptual model to guide research. This process began as a brainstorming activity and a review of the previously published RFE conceptual models. We agreed that there were elements of the RFE missing from previous conceptualizations. To address this, we began meeting throughout the next year, and with feedback from the wider membership, developed a conceptual model to reflect RFE evolutions and its complexity, as well as what has been learned about the RFE over the past 20 years of public health research. Our focus was on developing a model that captured the chronic, ongoing processes, and outcomes of the RFE, and much of our efforts preceded the recent COVID-19 pandemic and historic protests against police brutality across the U.S. While we believe some model components and outcomes are highlighted by the COVID-19 pandemic and the movement for racial justice, there are others we do not address (e.g., state-mandated restaurant closures). As a compliment, Leone and colleagues (see Special Issue “Retail Strategies to Support Healthy Eating” https://www.mdpi.com/1660-4601/17/20/7397) offer ways that the proposed model could be used to inform research directions during significant disruptions, such as pandemics.

The aim of this paper is to propose an updated conceptual model of the RFE and its relationships with customer behavior that produce a host of significant population outcomes. Below, we present an overview of the conceptual model and our underlying assumptions and motivations. We then describe and justify each of the model components. Last, we discuss how the model can be used to direct broad future directions in observational, intervention, and policy research to understand and modify the interactions between customers and the RFE with the intention of improving societal outcomes.

## 2. Overview and Motivation for the Retail Food Environment and Customer Interaction Model

As an overview, the Retail Food Environment and Customer Interaction Model (Figure 1) breaks the RFE down into business models, actors, and sources and their influence on the customer retail experience (e.g., food availability, promotion, quality). Our model depicts reciprocal relationships and influence between the RFE and customers, including their individual, interpersonal, and household characteristics that affect sales/purchases. The model highlights the multilevel context in which the RFE and customers operate and expands the population outcomes produced by RFEs and customers that should be considered moving forward: health, food security, food justice, environmental sustainability, and business sustainability. See Table 1 for component definitions.

Several underlying assumptions motivated the proposed model components and relationships. First, we took a highly-inclusive posture to address the multifactorial nature of the RFE in the US and its wide-ranging, discipline-crossing implications for society. However, we recognize as predominantly public health scholars that our focus remains on health and thus describe much of the model from that evidence base. We also conceptualized the model’s diverse and multidimensional components as a complex dynamic system. This is represented not only in the reciprocal relationship between RFEs and customers, but also by the inclusion of multilevel contexts that can affect RFEs, customers, and their interactions. Finally, we speculated that an important driver of the RFE evolution has been the supply and demand for convenience and highlighted this in several model components. Time scarcity [10,11,12], growing mental fatigue and stress [16], and changing social norms [17] around daily food preparation may all contribute to customers’ increasing demand for highly-accessible, limited-preparation products [18]. This demand has often been met by RFEs providing an abundance of ultra-processed, highly palatable, calorically-dense products through an ever-growing accessibility [19,20,21]. Yet, these patterns are juxtaposed by others that suggest that large swaths of the US are devoid of a variety of convenient foods and sources [22,23,24]. Thus, even an important driver, such as convenience, must be considered in a larger system of relationships and factors in order to understand why diverse outcomes can be produced.

## 3. Retail Food Environment

A key focus of our efforts to advance prior conceptualizations was to more comprehensively identify specific components of the RFE. We define the RFE as the environment where all food and beverages are purchased by consumers, including foodservice operations such as restaurants. We also recognize that the RFE is part of a larger food system, including agriculture, farming, and food production. However, in our model, we focus on the retail components most immediate to where food is sourced and purchased by customers, including: Retail Food Sources, Retail Food Actors, Retail Food Business Models, and the Retail Food Customer Experience. While differentiating the various components of the food environment is helpful, we acknowledge that overlap can and does exist among these components. 

### 3.1. Retail Food Sources

Retail food sources (e.g., stores, restaurants, websites) are settings where people can purchase food and beverages, and are a well-known, well-studied concept in food environment research. Most investigations have studied these sources by examining the geographic-related aspects, such as number of, proximity to, and density of food outlets (i.e., the physical locations whose primary business is to sell food, such as restaurants and stores) [13]. Using these measures, research has aimed to characterize community food environments and examine their associations with community residents’ diet and health related outcomes [25,26,27,28]. For example, prior evidence suggests positive relationships between convenience store availability and obesity among children [25] and between relative availability of unhealthy (e.g., fast food, convenience stores) to healthy (e.g., supermarkets, farmers’ markets) sources with adult obesity [28].

Yet, such conceptualizations of retail food sources have insufficiently addressed the full and evolving range of settings and modalities where food and beverages can be purchased. Business responses to address customer convenience (i.e., reduce customer time and effort in food preparation and acquisition) have likely driven a growth in retail food sources in the US [29] and contributed to an ever-increasing ubiquity of ready-to-eat foods and beverages available for purchase. Thus, our conceptualization (Figure 2) aims to capture a more complete range of retail food sources that have evolved and classifies them across two dimensions of customer convenience: food preparation and accessibility.

The first convenience dimension—food preparation—demonstrates the variation across sources in the typical proportion of products offered that are prepared: ready-to-eat versus unprepared. As shown in Figure 2, there is an apparent imbalance in the types of sources that primarily offer products that eliminate at-home food preparation versus those that offer unprepared versions. Some sources, such as fast food, restaurants, and food trucks, only offer ready-to-eat products. However, ready-to-eat foods are also staples in gas-marts and convenience stores through offerings of pre-packaged foods and increasingly grab-and-go delis and hot prepared food [30]. Even grocery stores and supermarkets are part of this prepared food trend [7,31], though continue to offer a greater percentage of products that require some (e.g., frozen pizza) or complete (e.g., eggs) at-home preparation. These offerings stand in contrast to other sources, such as farmer’s markets and meal kit deliveries, which continue to sell a majority of products that require some degree of preparation (e.g., cut, chop, and sauté fresh vegetables).

Sources have also evolved to address customer convenience through the dimension of accessibility. We view accessibility as the ability of customers to purchase products from their immediate location (e.g., home, work, school). Changes in accessibility were first observed through the staggering spread of brick-and-mortar food sources that narrowed customers’ travel distances to venues. For instance, evidence suggests that the density of fast food chains and restaurants near US homes and workplaces significantly increased between 1971 and 2008—in some cases doubling [32]. While these changes contributed to today’s approximately 200,000 fast food venues [33] and 153,000 convenience stores/gas-marts [34], accessibility has also recently evolved to no longer require people to travel to and visit brick-and-mortar locations. Such immediate accessibility has in some respects been around for decades through vending machines, worksite cafeterias, and pizza delivery. However, accessibility in recent years seems to be exponentially expanding. Ready-to-eat packaged foods (e.g., candy) are offered in non-food outlets and checkout aisles (e.g., barber shops, home improvement stores, clothing stores) [3,4]; sit-down and fast food restaurants regularly offer options for delivery, often via third-party online applications and platforms [9]; and even sources that primarily sell products requiring preparation are now delivering (e.g., meal kit deliveries, online grocery delivery). Moving forward, we need a better understanding of the impacts of these increasing forms of accessibility and prepared food products offered by retail sources. Future research can investigate how some modalities might be used to improve the ubiquity of healthier ready-to-eat options as well as disentangle for whom these convenience dimensions are more or less available.

### 3.2. Retail Food Actors

Retail food actors are the people that work in the RFE whom, at various steps in the process typically towards the middle and ends of the food supply chain, determine the foods and beverages available at a source (e.g., managers/owners, suppliers/distributors, merchandising managers, and sales representatives). The retail food actors interact to determine which items are feasible to sell, store, and transport while maintaining quality and minimizing waste. For instance, when source managers or restaurant owners plan to sell a new item, they identify potential suppliers and understand the space, cost, and shelf life requirements necessary to sell the product in a safe and profitable way. Food manufacturer sales representatives are another example, who work with store managers to promote products and marketing strategies, such as in-store displays [35].

Each actor has their own specialty and focus. A sales representative’s focus is often to develop relationships with retail outlets that will provide environments for food products to reach consumers and cultivate demand. A distributor’s focus may be to develop a supply chain that efficiently moves food from warehouses to stores and restaurants. Alternatively, a store manager or restaurant owner’s focus may be to provide an array of items that customers demand in an efficient and pleasant environment [36,37]. The varying foci and goals of these actors have often resulted in an efficient system that provides an abundance of convenient, non-perishable, manufactured food and beverage items, as these are often more logistically and financially appealing to manage [38,39,40].

Relative to other RFE components in the model, very little literature in public health nutrition has investigated the impact of these actors on the RFE, though there is a growing base of research examining the role of store managers [36,41,42,43]. Such research is important as these actors develop reciprocal and deterministic processes that influence the current RFE (e.g., informal and formal product contract agreements [44,45], managers requesting products from distributors based on customer demand and what they can maintain due to resources and infrastructure) [46,47]. A better understanding of how the retail food source is influenced by the goals, foci, and decisions of these actors may be necessary to develop more effective policies and sustainable interventions to improve population outcomes.

### 3.3. Retail Food Business Models

Another RFE component that requires additional research is the business models used across each retail food source. Business models direct a source’s operations, financing, target customer base, and mission. Understanding the business model of a source, particularly products sold and services provided, helps to understand their priorities. For example, sources offering culturally-tailored products might be demonstrating a priority to address the needs and preferences of a specific ethnic community [48,49,50], while sources offering products with specific values, such as locally-sourced, or dietary requirements, such as gluten-free [51], may be targeting and prioritizing other customer groups. Services provided (e.g., fast food versus “dining experience”) can also indicate a source’s targeted customer base (e.g., income/class, available time, cooking abilities/preferences). Products sold might also reflect a source’s priorities to generate additional revenue streams, such as stores that participate in federal assistance nutrition programs [52] to expand their customer base, as well as how much local demand is valued over operational convenience (e.g., product variation versus the same products at all locations) [53].

Business outcomes, including revenue and profits, are often the ultimate goal for many sources. Such goals are at times a necessity, given that some source types (e.g., grocery wholesalers and stores) struggle with low profit margins [54]. Yet, some sources may have additional goals beyond profit. For example, institutional foodservice companies might be profit driven, but they contract with community-based institutions, such as colleges, workplaces, prisons, or hospitals [55]. This partnership creates a mix of profit motive and community benefit where the institution’s goals, such as for healthy eating and/or locally-sourced products, influences the foods that the foodservice company provides.

Ownership is another indicator of the business model, and a range of ownership types with diverse goals exist across the RFE. The majority of foods and beverages purchased in US are sold by publicly-traded corporations [56,57,58], such as Walmart, Kroger, McDonalds, Sysco, and Starbucks. However, there are a number of large-chain food retailers that are privately-owned such as Chick-fil-A, Publix, Meijer, and Subway. Such privately-owned chains, while not always held to produce profits for shareholders, continue to dominate RFE spaces (in terms of profits, reach, etc.) over the private and independently-owned source with only one or two locations. Other examples of ownership models include food cooperatives (co-ops) and community-owned businesses. Co-ops involve groups of people that use membership fees to collectively operate a food retailer. Some co-ops are not-for-profit companies, allowing more flexibility to operate the co-op in a manner aligning with the co-op’s mission or changing member needs. Community-owned business food retailers are often for-profit businesses that are financed, owned [59], and operated collectively by community members (e.g., Baldwin Market in Florida), and differing from co-ops often raise more capital and investments to allow “capital-intensive enterprises to start at scale [59]”.

The past several decades have brought an important RFE transformation from small independent ownership to large chain often corporate/franchise ownership [60,61,62]. In some cases, entire groups of sources may be corporately-owned, such as fast food. In other cases, ownership at sources, such as grocery stores and supermarkets, remains relatively diverse; though, these also show growing declines in the presence (number and market share) of independent ownership [60]. With these shifts in centralizing ownership to fewer hands, much remains to be investigated and understood about how these different ownership types and business models contribute to the RFE [63,64].

### 3.4. Customer Retail Experience

Together, retail actors, business models, and retail sources combine and lead to the final component of the RFE: the customer retail experience. This component consists of the characteristics of food and beverage products for sale and the broader environment that people encounter when making their purchases. Referred to by Glanz and colleagues as the “consumer nutrition environment” [13], these features were mainly conceptualized as occurring within a physical location. Yet, given increasing shifts to online purchasing, customers are now also experiencing retail food spaces through webpages and mobile applications.

The traditional marketing mix of product, price, place, promotion, and people remains a helpful way to classify the customer retail experience [65]. In comparison to research on retail food sources, fewer studies in the field have examined how features of the customer retail experience within those sources relate to purchasing, consumption, or health outcomes [26,27]. This work is important as studies examining links with sources often rely on classifying entire source types as either healthy (e.g., supermarkets) or unhealthy (e.g., fast food); yet, this can neglect the variation in product mixes (e.g., supermarkets offer plenty of unhealthy products), placement, and other marketing features within a source that influence customer purchasing [66,67,68].

Of the limited evidence examining features of the customer retail experience, many have studied food product availability or prices. Both the absolute and relative availability and prices of healthful and unhealthy foods, as well as availability of culturally-appropriate products [49,50], may be relevant for consumers’ purchasing decisions [69,70,71]. Often, unhealthy products are more available [72,73,74,75] and less expensive than healthful products [76]. Product quality and variety (i.e., number of options), such as for produce or milk options, also influence purchasing decisions [77,78,79,80,81] and can vary across source type and neighborhood [82,83,84].

Other features, including placement and promotion, have been less studied, although industry practices provide indirect evidence that these, too, are important for creating a customer retail experience that translates into sales. For instance, food/beverage manufacturers spend an estimated USD 50 billion per year, or 70% of their marketing budget, on in-store trade-promotion fees [35]. Such fees can guarantee certain product placement (e.g., checkout aisles) and/or promotion through cooperative advertising (e.g., store circulars) and discount campaigns (e.g., “2 for 1”). These practices also occur in online shopping spaces, such as pop-up advertisements, notifications, and cart “reminders” [85]. Promotion also occurs at the packaging level, as significant efforts have been made by manufacturers to attract customers (e.g., children’s cereal boxes [86]) and by public health to inform customers of a product’s nutritional composition and quality (e.g., nutrition label reform [87], front-of-package, and traffic-light labeling [88,89,90]). Even newer features of shelf promotion, such as undershelf lighting in the candy aisle, signals that these features will continue to evolve as the competition for customer attention and thus sales endure among companies and product categories [91].

The final feature—people—also affects customers’ decisions on where to shop and the food and beverage products to which they have access. Despite limited literature, studies indicate that negative social interactions influence people’s shopping locations and can range from inefficient, unenthusiastic service to forms of discrimination and stigma [92,93,94,95]. For instance, Black Americans have described employees watching, following, or treating them with less respect and experienced this behavior while shopping in predominantly White neighborhoods or in stores owned by individuals of a race/ethnicity different than their own [94,96,97,98,99]. Research also highlights that some customers frequent sources that they trust and especially those with which they have a built relationship [100,101].

## 4. Retail Sales and Customer Purchasing

The conceptual model involves two sides—an RFE side that presents key components that are most immediate to where food is sourced and purchased by customers and a customer side that presents the many aspects relevant to individual variation in customer purchasing and dietary intake (see Section 5). The two sides connect at the point of a transaction or where a product is sold by the retailer and equivalently purchased by the customer (Figure 1). In comparison to prior models, we view the relationship between the two sides as reciprocal: actions of RFEs influence customer behaviors and customer behaviors influence RFE actions. Moreover, aspects of both the RFE and individual customers can interact to lead to a customer purchase.

## 5. The Customer: Individual Dietary Intake, Individual Characteristics, and Household Characteristics

The right-side of the model (Figure 1) represents the customer and the multidimensional characteristics that influence decisions about personal dietary intake and food and beverage purchasing. The relationship between dietary intake and purchasing is bidirectional, and we propose that a wide range of individual, interpersonal, and household characteristics influence individual purchasing and ultimately dietary intake (Table 2) [14]. At the individual level are the intrapersonal factors that influence dietary intake and purchasing behaviors. Previous models and a large body of evidence indicate that factors such as attitudes, knowledge, food preferences, socio-demographic characteristics, lifestyle behaviors (e.g., smoking), stress, and cultural norms influence these behaviors. For example, individuals who have less education and/or poor employment consistently report lower dietary quality [102,103], which may be due to limited time or financial resources. Individuals’ food knowledge and attitudes are also important, as greater nutrition knowledge has been associated with better dietary quality and may reflect a better nutrition label literacy and ability to overcome food marketing tactics to make healthier purchases [104].

Individuals are embedded in households and other interpersonal contexts, and much evidence suggests that characteristics from these contexts, such as SNAP status, income, social support, social norms, shopping behaviors, and food preparation skills and decision-making, are also related to dietary intake and purchasing. For example, sources and timing of food benefits (e.g., SNAP benefit schedule) shape the number of food shopping trips and their food baskets, as diets tend to be healthier around the weeks that SNAP household benefits are redeemed versus other times of the month [105,106]. Household income is also consistently related to purchases with higher household incomes purchasing healthier foods and beverages and having greater access (e.g., fruits and vegetables) compared to lower income households [107]. In households of immigrant families, the level of acculturation of the head of household influences what food is purchased [108,109]. Additionally, household members’ work schedules and transportation options are related to shopping trip frequency and foods purchased and may contribute to customers’ increasing need for convenience [110].

## 6. Community, State, Tribal, National, and Global Contexts

As shown in Figure 1, the RFE, as well as customers do not operate in a vacuum. Instead, macro-level factors, including economic, social, media, built environment, policy, and others, influence the RFE, customers, and their relationships. A growing body of evidence examines factors at this macro-level, and we group these factors under two contextual levels: community context, including neighborhoods and city/local jurisdictions, and the broader state, tribal, national, and global context. Table 3 provides examples of relevant factors in each context. These factors may directly affect the RFE and customers, as well as modify RFE–dietary intake relationships.

Under the community context, zoning codes and commercial real estate professionals can directly influence where food sources are located; transportation systems can affect the food sources that consumers can reach; local social norms might ultimately influence the food options available; and tax policies such as municipal sugar-sweetened beverage taxes affect the prices that consumers face [111]. Under the broader contexts, numerous factors from the state, tribal, and federal levels influence food retail and customer behavior such as the following examples. First, stocking requirements for participating retailers in SNAP and the Special Supplemental Nutrition Program for Women, Infants, and Children (WIC) affect the products they carry, and these are particularly impactful in smaller sources, such as dollar stores, that may not otherwise carry as many healthier items [43,112]. Second, minimum wage laws affect the financial resources that consumers have to purchase foods. Third, industry advertising of food products may affect food choices and thus dietary intake. Finally, healthy food purchasing incentive programs and nutrition education programs (e.g., within SNAP) increase financial access and education about healthy foods and beverages. As suggested by the embedding of the community context within the broader context in Figure 1, factors at the state, tribal, national, and global levels can influence the community context as well. For example, preemptive laws can prevent state and local governments from enacting policies that influence the RFE, such as taxes and labeling [113].

Moreover, community and broader contexts may modify effects of the RFE on sales/purchasing and in turn individual dietary intake. For instance, the effect of retail food outlet availability on where people purchase foods may depend on safety of the surrounding community environment, as shoppers have reported avoiding stores or certain shopping times due to unsafe neighborhood conditions, including drug sales, violence, and harassment [93,94]. Because consumption of high-fat, sugary foods and beverages can alleviate stress, exposure to such community stressors may also increase the effect of household availability of these food types on individual consumption choices. With regard to the broader state, tribal, national, or federal context, for example, the impact of in-store food/beverage marketing to children on purchasing may vary depending on regulations for television food advertising to children. That is, it is possible that being exposed to both in-store marketing and television food advertising will have a stronger impact on purchasing than when one is only exposed to one or the other. Thus, our model recognizes a wide variety of factors that may influence the RFEs, customers, and their sales/purchasing interface as well as alter these relationships.

## 7. Population Outcomes

We posit that the dynamics and interactions between RFEs, individuals and households, and their larger contexts can produce a host of population outcomes. Scholars have previously articulated the importance of examining the multiple outcomes produced by national and global food systems [114,115]. In this conceptualization, we offer five for consideration: health; food security; environmental sustainability; business sustainability; and food sovereignty, equity, and justice.

Population health is the outcome most familiar to RFE researchers from the discipline of public health. It aims to uncover the ways this system contributes to diet-related non-communicable diseases, such as obesity, type 2 diabetes, and cardiovascular disease [13,14]. Those interested in improving this outcome often offer RFE modifications that help “make the healthy choice, the easy choice,” such as offering and widely promoting products consistent with national dietary guidelines [116].

Food insecurity is another outcome, and one that at the time of this writing has dramatically risen as a result of the economic implications from executive orders required to curb the spread of COVID19 (e.g., miles of cars waiting at food pantries [117]) as well as damage to RFE locations that accompanied peaceful protests for racial justice [118]. In usual times, food insecurity is likely to occur when federal nutrition assistance is not accepted at all sources, price structures lead high-fat and high-sugar products to be most affordable (i.e., lowest-cost dietary option), and fresh and healthier options are not equally available across communities. As such, healthy food and beverages are not affordable or accessible to all groups, and this most often impacts the economically and socially disadvantaged [14,116].

Outcomes of the system not only relate to people but to the environment. Food waste is one example, as more than 400 pounds of approximate waste per person was observed at the US retailer and consumer levels in 2010 [119]. Other examples relate to the agriculture and transportation practices required for the types of products sold and purchased. Many suggest that the majority of available products are produced and commonly transported in a way that leads to environmental degradation, as they require methods that can diminish soil fertility, emit greenhouse gases, deplete freshwater resources, and/or neglect biodiversity [114,115].

The system also contributes to business and economic outcomes, which reflect the “health” of the source’s business performance in the US market economy [120]. Here, goals of generating sales, profits, and competitiveness are key and for some businesses may be the primary motivators for decision-making [120,121,122,123]. Food retailers and companies often aim to achieve such goals by interrogating consumer “choice” and the predictors of which retailers will be shopped and which products purchased (e.g., price strategies, product mix, store layout) [122,124]. Of the outcomes identified, this outcome has arguably been the best performing in recent decades, as US supermarket and fast food industries experienced an estimated annual revenue in 2019 of USD 682 billion and 293 billion [125,126], respectively. However, the COVID-19 pandemic is likely to change this success for some industries, such as restaurants, which observed a 51% drop in food-away-from-home monthly expenditures in March 2020 compared to March 2019 [127].

Finally, there are also significant outcomes of the system characterized through the lens of food sovereignty, equity, and justice. In this perspective, inequalities in power are central, and the rights of individuals and communities to define, produce, and sell their own food are emphasized [116,128]. To achieve such outcomes requires addressing the socio-structural barriers (e.g., economic inequality, racism, sexism) that have historically-marginalized, inequitably targeted, and resource-deprived certain groups and populations [129]. It also demands the development of sustained opportunities for communities to create the RFE that best serves their needs and interests (e.g., supporting tribal food sovereignty and Black-owned businesses).

Articulating these five outcomes is necessary to not only highlight the multiple outcomes produced and that need to be considered in future research but the challenges and opportunities that also lie ahead. For instance, when we focus on a single RFE goal and ignore that other outcomes are produced, we create solutions that may address our goal but simultaneously produce harm in other areas. Such consequences may be unintentional or well-known (e.g., promotion of unhealthy, processed foods which increase profits, but are associated with non-communicable disease [130,131,132]). Yet, moving forward it may be important to reframe these varying outcomes from inevitable systemic trade-offs to sites of opportunity. Diverse groups working to improve the RFE could identify ways to work at cross-purposes, achieve goals for multiple outcome areas, and potentially do so with greater efficiency and less duplication and resources. Working together will also push discipline-centric change agents to consider the feasibility and sustainability of their proposed solutions and may help spur the creation of more worthwhile and effective transformations. While collaboration and attention to multiple outcomes will be easy for some, other groups may require support or even accountability measures to help cultivate “common ground” (e.g., reframing from businesses profitability to sustainability), and many have already been calling for and provided specific strategies to do so [115,129,131,133].

## 8. Future Directions

The Retail Food Environment and Customer Interaction Model attempts to capture key RFE and customer components in the US that converge to shape food and beverage purchases with diverse societal outcomes. Expanding upon previous frameworks [13,14,15], we believe this updated model highlights: (1) the multifactorial nature of the RFE; (2) the wide-ranging and discipline-crossing outcomes produced for society; (3) the reciprocal and dynamic relationships between RFEs and customers as well as with factors from multilevel contexts creating a complex system; and (4) the importance supply and demand for convenience has and continues to play in shaping the US RFE. As such, the model adds important information that can guide future research on the broader RFE context for dietary intake and help to inform public health interventions and policies aimed at improving RFE settings.

The encompassing nature of our model has broad implications for future research and can guide numerous research questions. However, here for the sake of brevity, we focus our comments on three important gaps that we identified throughout model development. First, additional research is necessary to investigate the role and influence of certain understudied RFE components: retail actors, business models, and the customer retail experience. A better understanding of these components is required to develop effective interventions and partnerships that are more likely to improve outcomes. Second, there is a need, especially in public health, to broaden our awareness of outcomes beyond health in an attempt to anticipate the wide array outcomes that a single change to the RFE and customer interaction can generate. Finally, while literature examining why convenience is an important driver of behavior exists in the disciplines of psychology, behavioral economics, and cognitive science, there remain relatively less investigation and understanding of nutrition and public health. Uncovering what convenience means to customers and how best to capitalize on it to improve health and other population outcomes are important directions moving forward.

Given the complexity, dynamics, and reciprocal processes of the Retail Food Environment and Customer Interaction Model, we also suggest a need for more sophisticated research methods and transdisciplinary partnerships. Two recommended research approaches are systems science and multilevel, multicomponent (MLMC) interventions [134,135,136]. Systems science involves methodological approaches, often computational models, that aim to understand the impacts produced from complex interrelated mechanisms and relationships among multiple factors [134,135]. Except for a few exceptions [137,138,139,140], relatively little work has studied the RFE using such methods, and incorporating these approaches could help to not only identify solutions that improve multiple outcomes but identify those to avoid to circumvent unexpected consequences. MLMC interventions are large, complex, multidimensional interventions that often require significant coordination, stakeholder buy-in, and resources; yet, their utility also lies in identifying which individual and/or set of components most effectively improves outcomes [141,142]. Both the model’s complexity and these research approaches suggest that transdisciplinary, collaborative leadership will be required. Bringing together stakeholders from many disciplines, such as agriculture, business, public policy, regional/urban planning, nutrition, social sciences, and public health, could help to build more and stronger transdisciplinary projects that are better positioned to effectively improve the RFE for a variety of societal outcomes.

## 9. Conclusions

This paper provides a model depicting the interactions of the RFE and consumer behavior while also highlighting some of the outcomes of this system as witnessed in the US. We view the Retail Food Environment and Customer Interaction Model as a “living” conceptualization and hope that it inspires many additional, more refined versions. We encourage research utilizing this model to help us better understand why food sources operate in certain locations, how food sources decide which foods to carry, and why customers choose to purchase certain foods. Then using this insight, transdisciplinary efforts should work to develop solutions that modify the RFE-customer relationship in ways that ultimately improve a range of population outcomes. 

## Figures and Tables

**Figure 1 ijerph-17-07591-f001:**
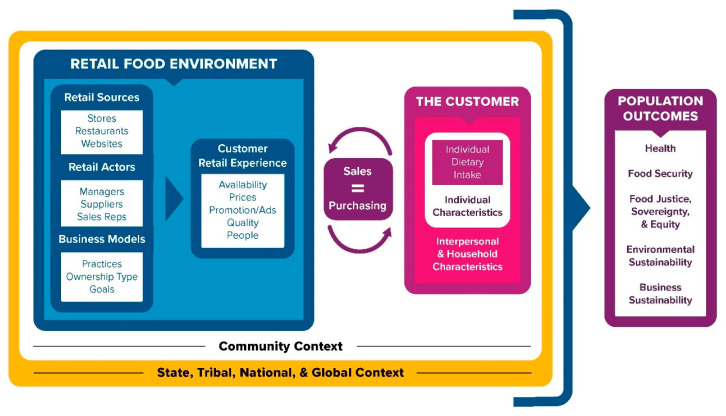
Retail Food Environment and Customer Interaction Model. The retail food environment consists of retail sources, retail actors, and business models that influence the customer retail experience. Customers involve individual, interpersonal, and household characteristics that affect customer purchasing and thus the retail sales of foods and beverages. Both the retail food environment and customers and their households are embedded in macro-level contexts (e.g., communities and nations), and as a result of the interactions and dynamics among these multiple model components, a host of population outcomes are produced: health, food security, food justice, environmental sustainability, and business sustainability. Definitions for model components are provided in Table 1.

**Figure 2 ijerph-17-07591-f002:**
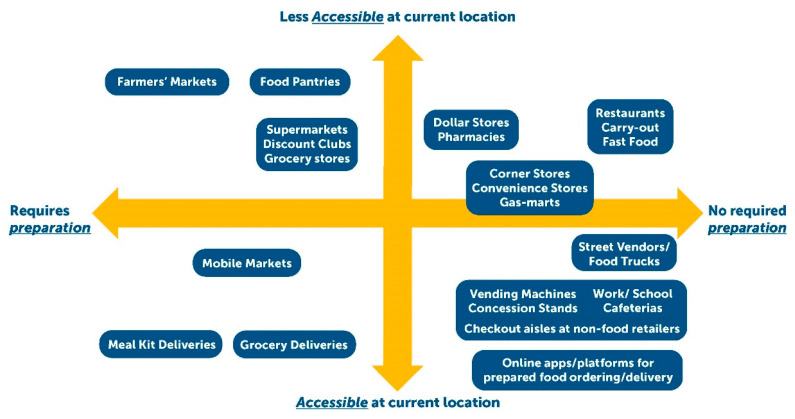
Common and Emerging Retail Food Sources across Two Dimensions of Customer Convenience: Accessibility and Degree of Required Preparation. Accessibility involves the ability for customers to obtain food products from a retail source from their immediate location (e.g., food can be delivered to their location or customers are required to travel to source). Degree of required preparation captures the typical proportion of products offered by the source that is prepared: ready-to-eat versus unprepared.

**Table 1 ijerph-17-07591-t001:** Definitions for key model components.

**Retail Food Environment**	
Sources	• The settings (e.g., stores, restaurants, online websites/apps) where people can purchase and obtain food/beverage products
Actors	• The people who interact, make decisions, and behave in various ways that create and support the current food environment, such as: store managers, owners, distributors, wholesalers, and sales representatives
Business Models	• The business design (e.g., targeted customer base, product/service selection), practices, goals, and ownership types (e.g., independent, publicly-traded, franchise) that characterize retail food businesses
Customer Retail Experience	• The features (e.g., price, availability) that customers encounter when they obtain and purchase food/beverage products
**The Customer**	
Individual Dietary Intake	• The specific foods and beverages consumed
Individual Characteristics	• Factors at an intrapersonal level that contribute and influence individual dietary intake and/or purchasing behavior
Interpersonal and Household Characteristics	• Factors at the interpersonal and household levels that contribute to an individual’s behavior and characteristics
**Sales and Purchasing**	• The point of a transaction where a product is sold by the retailer and equivalently purchased by the customer
**Community Context**	• Macro-level factors from neighborhoods and city/local jurisdictions that influence the retail food environment, customers, and their relationships.
**State, Tribal, National, and Global Context**	• Macro-level factors from state, tribal, national, and global contexts that can influence the community context, retail food environment, customers, and their relationships.

**Table 2 ijerph-17-07591-t002:** Examples of individual, interpersonal, and household characteristics relevant to food and beverage purchasing and dietary intake behavior.

Domain	Individual Characteristics	Interpersonal and Household Characteristics
**Examples**	Eating behaviors Food cooking skills and behaviors Taste/food preference/meal selection Cognitions (attitudes, knowledge, preferences) Time availability and pressure Perceived stress and physiologic stress responses Lifestyle/other health behaviors Weight status Eating disorders and chronic health conditions Biological (age, genes) Demographics (education, race/ethnicity, employment) Immigration status Cultural values Prior experiences/memories with food	Household membership Food preparation equipment, tools, and space Household member with food preparation skills Work schedules Transportation US Department of Agriculture Supplemental Nutrition Assistance Program (SNAP) status Time of month (food benefit cycle) Household preferences for food/drinks available Social influences (role modeling, support, norms) Food purchase frequency Time of the day Access to and placement of foods in the home Food choice incentives Rules and norms about eating (family eats together)

**Table 3 ijerph-17-07591-t003:** Macro-level contexts with example factors that influence the retail food environment; customer purchasing, dietary intake, and individual and household characteristics; and their relationships.

	Retail Food Environment	Customer: Diets and Individual and Household Characteristics	Community Context
**Community Context**	Licensing fees Taxes (e.g., sweetened beverage taxes) Local subsidies Income level and purchasing power Cost of living Local ordinances (e.g., default beverage in restaurant child meals, staple foods) Food industry contracts with schools, hospitals, and other institutions Zoning codes	Economic development Employment opportunities Safety Retailer-community relations Social and cultural norms Stressors (e.g., disorder, violence) Educational system Transportation systems Walkability Public health campaigns Food industry sponsorship of community activities (e.g., child sports/summer camps)	
**State, Tribal, National, and Global Context**	Food assistance programs–retailer requirements (e.g., SNAP, WIC) Banking and lending practices Societal values and ideologies Broadband internet infrastructure School, daycare, worksite policies Regional planning Food safety standards Food labeling laws Food production Product development Food processing/manufacturing Marketing (e.g., trade promotion fees) Agriculture policies and subsidies International trade agreements	Federal nutrition assistance programs–benefits and food packages (e.g., SNAP, WIC) Minimum wage laws Regulations for media advertising to children Advertising (e.g., commercials, social media, sponsorships)	Funding for education Transportation funding Preemption laws

Note. SNAP, US Department of Agriculture Supplemental Nutrition Assistance Program; WIC, US Department of Agriculture Special Supplemental Nutrition Program for Women, Infants, and Children.

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
