# Peer review of "A Model Depicting the Retail Food Environment and Customer Interactions: Components, Outcomes, and Future Directions"

_ijerph, 2020, doi:10.3390/ijerph17207591_

Round 1

Reviewer 1 Report

The authors provide a very interesting model to study the food retail environment more broadly. The model is encompassing, which allows to study the multidimensionality of food consumption. As the authors identify and discuss the different dimensions of their model, it is not always clear for the reader when they are building on existing research and where they rather see a blind spot in the literature. I think this could be improved and would set more clearly a research agenda around the core model that they propose.

Author Response

The authors provide a very interesting model to study the food retail environment more broadly. The model is encompassing, which allows to study the multidimensionality of food consumption.

  • Thank you for this kind feedback.

As the authors identify and discuss the different dimensions of their model, it is not always clear for the reader when they are building on existing research and where they rather see a blind spot in the literature. I think this could be improved and would set more clearly a research agenda around the core model that they propose.

  • Thank you for these comments. We incorporated edits throughout the article to better clarify model aspects that build on existing empirical work versus those included which currently are considered literature gaps.
  • For example, we now state: “Relative to other RFE components in the model, very little literature in public health nutrition has investigated the impact of these actors on the RFE, though there is a growing base of research examining the role of store managers [36,41-43]” (lines 222-224); and “A better understanding of how the retail food source is influenced by the goals, foci, and decisions of these actors may be necessary to develop more effective policies and sustainable interventions to improve population outcomes.” (lines 227-230).
  • If you have additional suggestions to improve these edits, please do not hesitate to let us know.

Reviewer 2 Report

A Model Depicting the Retail Food Environment and Customer Interactions: Components, Outcomes, and Future Directions

The paper presents a model that has played an important role in research. The contribution of the present study is the opportunity to examine the retail food environment with some components such as actors, sources... This paper could be relevant in public health and primary prevention about dietary habits in the moment of purchase. However, the manuscript needs a revision in order to improve the Introduction, and other sections.

A series of major revisions are needed.

Introduction:

The introduction in the present form seems a bit confusing. To improve the fluency of the first paragraph could be modified as follows: introduction about the the potential consequences of malnutrition, published reviews accounting for the similar models published previously. The contribution of this models in the nutrition research.

Lines 68-84. The authors should describe the previous evidence about the results of this models in general. The authors may added the comparison with other populations models and with results obtained by other authors but they should not describe their research group, this is not really important in the introduction of the topic.”

Lines 85-86 “Our efforts in developing this conceptual model largely preceded the COVID-19 pandemic and historic protest against the police brutality”. I really feel that this phrase is not relevant for the article.

Finally, the authors have to highlight the main aim of the present study.

Retail Food Sources

Lines 158-164. Authors should describe how different kind of retail food sources can affect health, e.g: fast-food restaurants vs supermarket.

Customer Retail experiece

Authors have also to explain how the composition of food can influence purchasing decisions (sugar,fat, fiber…) . Some customers can use algo the nutritional traffic light.

Future directions

The authors should discuss why this model is important for nutritional research.

What is the novelty of this paper?

Author Response

The paper presents a model that has played an important role in research. The contribution of the present study is the opportunity to examine the retail food environment with some components such as actors, sources... This paper could be relevant in public health and primary prevention about dietary habits in the moment of purchase.

  • Thank you for these comments. We also hope the model helps to guide future investigators in prevention science and public health nutrition.

However, the manuscript needs a revision in order to improve the Introduction, and other sections. A series of major revisions are needed.

Introduction:

  • The introduction in the present form seems a bit confusing. To improve the fluency of the first paragraph could be modified as follows: introduction about the potential consequences of malnutrition, published reviews accounting for the similar models published previously. The contribution of this models in the nutrition research.
  • Lines 68-84. The authors should describe the previous evidence about the results of this models in general. The authors may added the comparison with other populations models and with results obtained by other authors but they should not describe their research group, this is not really important in the introduction of the topic.”
    • We appreciate the suggestions for the introduction, including re-ordering information in the first paragraph and removing information about the author group and process used to develop this model. Rather than making all the suggested changes, we have chosen to also prioritize the feedback of reviewers 1, 3, and 4 who found the introduction to provide a sufficient background and in particular commented that “the methodology was presented clearly.”
    • In response to your suggestions, we have edited the paragraphs (original lines 68-84) about the author group to clarify the background specific to the working group itself from the timing and process used in developing the model.
    • Instead of removing the content about our methods and workgroup, we are open to moving it to section 2 and expanding that section to be a ‘model overview and methodology’ section, if the editor deems these changes are necessary given other reviewer feedback.

  • Lines 85-86 “Our efforts in developing this conceptual model largely preceded the COVID-19 pandemic and historic protest against the police brutality”. I really feel that this phrase is not relevant for the article.
    • We edited the language and modified the sequence of sentences in original lines 85-86. We believe these edits more clearly specify the original goals in model development and its timing around more recent events (COVID-19 and protesting for racial justice across the US), which has had a significant impact the retail food environment in many of our own cities and work (lines 82-84).

  • Finally, the authors have to highlight the main aim of the present study.
    • We added a statement that articulates the aim of the manuscript: “The aim of this paper is to propose an updated conceptual model of the RFE and its relationships with customer behavior that produce a host of significant population outcomes” (lines 89-90).

Retail Food Sources

  • Lines 158-164. Authors should describe how different kind of retail food sources can affect health, e.g: fast-food restaurants vs supermarket.
    • We have expanded information in lines 151-156 to provide examples of prior research in the field that has observed relationships between specific food sources and health outcomes: “Using these measures, research has aimed to characterize community food environments and examine their associations with community residents' diet and health related outcomes [25-28]. For example, prior evidence suggests positive relationships between convenience store availability and obesity among children and between relative availability of unhealthy (e.g., fast food and convenience stores) to healthy (e.g., supermarkets and farmers’ markets) sources with adult obesity.”

Customer Retail experience

  • Authors have also to explain how the composition of food can influence purchasing decisions (sugar, fat, fiber…) . Some customers can use algo the nutritional traffic light.
    • Thank you for this suggestion. We have edited the information that discusses product labelling to ensure nutritional composition and other forms of nutrition labelling, including traffic light labeling, are acknowledged and included.
    • Lines 305-308: “Promotion also occurs at the packaging level, as significant efforts have been made by manufacturers to attract customers (e.g., children’s cereal boxes [86] ) and by public health to inform customers of a product's nutritional composition and quality (e.g., nutrition label reform [87], front-of-package and traffic-light labeling [88-90]).”

Future directions

  • The authors should discuss why this model is important for nutritional research.
  • What is the novelty of this paper?
    • Thank you for this feedback. We now clarify what is novel about the model and the research implications it has for the field.
    • “Expanding upon previous frameworks, we believe this updated model highlights: 1) the multifactorial nature of the RFE; 2) the wide-ranging and discipline-crossing outcomes produced for society; 3) the reciprocal and dynamic relationships between RFEs and customers as well as with factors from multi-level contexts creating a complex system; and 4) the importance supply and demand for convenience has and continues to play in shaping the US RFE. As such, the model adds important information that can guide future research on the broader RFE context for dietary intake and help to inform public health interventions and policies aimed at improving RFE settings.” (lines 473-480)
    • “The encompassing nature of our model has broad implications for future research and can guide numerous research questions. However, here for the sake of brevity, we focus our comments on three important gaps we identified throughout model development...” (lines 481-483)

Reviewer 3 Report

Thank you for an interesting article.

The methodology is presented clearly and the article is an easy read.

I hope that the comments in the review would be helpful:

  • I am asking for a clearer indication in the abstract about the overview nature of the work
  • Better photo quality Figure 2 and description below the figure as written text, not a photo
  • everywhere use the full name or abbreviation of tables / figures in the text consistently for table / figure headings

Author Response

Thank you for an interesting article.

The methodology is presented clearly and the article is an easy read.

  • Thank you for this supportive feedback.

I hope that the comments in the review would be helpful:

  • I am asking for a clearer indication in the abstract about the overview nature of the work
    • Thank you for bringing this to our attention. Working within the abstract word count restrictions of 200 words, we have revised the abstract to provide a clearer overview of the work/model (e.g., deleting less necessary information, re-ordering and/or adding information about the model). (lines 21-31)

  • Better photo quality Figure 2 and description below the figure as written text, not a photo
    • We have updated Figure 2 to improve the quality and added a figure legend to both Figures 1 and 2.
    • We are also including the figures as separate pdf files so that (if accepted) the IJERPH editorial team can help assist us and ensure that figures inserted into the final version are to the utmost quality for readers.

  • everywhere use the full name or abbreviation of tables / figures in the text consistently for table / figure headings
    • We have revised the use of this language throughout the manuscript to ensure we are consistently using the full name of Table and Figure.

Reviewer 4 Report

Very well done over-all. Minor editorial suggestions include:

line 48-49 more clearly define 'food purchased away from home' in the follow-up, it seems t specify already prepared food, but I would suggest that the majority of food, even at farmers markets, is purchase away from the home. Rather, if it is meant any food not ordered to the home, such as meal delivery kits, that should be clarified. 

line 54 the ,too is redundant to the preceding also

lines 65-66, usage rules vary, but I suggest commas rather than semi-colons here

Table 1: under customers, household- has a double space after and in sales and purchasing transaction has two hyphens double space, but this is not the convention throughout, so I suggest making uniform throughout the paper

Figure 2: while prepared foods can, a noted, be purchased at supermarkets, the graphic makes grocery store purchase less accessible but requiring more preparation than meal kits, which seems backwards- similarly, in the more accessible, more preparation quadrant, meal kits appear to require more preparation than at home grocery delivery- this again seems erroneous, of course depending on the product purchased.

line 451 I suggest  a colon rather than a hyphen

Author Response

Very well done over-all.

  • Thank you for this feedback.

Minor editorial suggestions include:

  • line 48-49 more clearly define 'food purchased away from home' in the follow-up, it seems to specify already prepared food, but I would suggest that the majority of food, even at farmers markets, is purchase away from the home. Rather, if it is meant any food not ordered to the home, such as meal delivery kits, that should be clarified. 
  • line 54 the, too is redundant to the preceding also
  • lines 65-66, usage rules vary, but I suggest commas rather than semi-colons here
  • Table 1: under customers, household- has a double space after and in sales and purchasing transaction has two hyphens double space, but this is not the convention throughout, so I suggest making uniform throughout the paper
  • line 451 I suggest a colon rather than a hyphen
    • Thank you. We have incorporated your suggested edits for all the above.

  • Figure 2: while prepared foods can, a noted, be purchased at supermarkets, the graphic makes grocery store purchase less accessible but requiring more preparation than meal kits, which seems backwards- similarly, in the more accessible, more preparation quadrant, meal kits appear to require more preparation than at home grocery delivery- this again seems erroneous, of course depending on the product purchased.
    • We appreciate these comments. We selected to place meal kit deliveries as the anchor example for sources that always require preparation and is most accessible to a specified location. This decision was based upon the kit deliveries we are aware of, which often include meal ingredients delivered directly to a person’s home and require some degree of preparation (e.g., chopping, heating, etc.). As the anchor, we then considered the typical proportion of products offered at the other sources that require preparation but perhaps not the same proportion as required for meal kit deliveries. We felt that sources, such as grocery stores and grocery deliveries, while certainly can include only preparation-required items (depending on what is purchased), also sell a larger proportion of already prepared products (e.g., snack cakes, chips) than obtained through meal kits; as such, we placed these to the right of meal kits to indicate that some products obtained from these sources (on average) will not require preparation.
    • We have modified the language about grocery stores to clarify their unique role in offering both prepared and unprepared products, and identify meal kit deliveries as an additional example of a source that primarily sells unprepared products: “Even grocery stores and supermarkets are part of this prepared food trend [7,31], though continue to offer a greater percentage of products that require some (e.g., frozen pizza) or complete (e.g., eggs) at-home preparation. These offerings stand in contrast to other sources, such as farmer's markets and meal kit deliveries, which continue to sell a majority of products that require some degree of preparation (e.g., cut, chop, and sauté fresh vegetables).” (lines 179-184).

Round 2

Reviewer 2 Report

The paper is well written and it is interesting. Thank you for the new version.